# Therapeutic Strategies of Biologics in Chronic Rhinosinusitis: Current Options and Future Targets

**DOI:** 10.3390/ijms23105523

**Published:** 2022-05-15

**Authors:** Junhu Tai, Munsoo Han, Tae Hoon Kim

**Affiliations:** Department of Otorhinolaryngology-Head & Neck Surgery, College of Medicine, Korea University, Seoul 02841, Korea; junhu69@korea.ac.kr (J.T.); mshan35@gmail.com (M.H.)

**Keywords:** chronic rhinosinusitis, biologics, therapeutic strategy, immunity, monoclonal antibody

## Abstract

Chronic rhinosinusitis is a chronic inflammatory disease of the upper airways, for which treatment options include medical or surgical therapy. However, there are limitations to conservative treatment strategies, such as the relapse of nasal polyps. In this review, we discuss the rising role of biomolecular mechanisms associated with various biologics that have been approved or are undergoing clinical trials to treat chronic rhinosinusitis. We also highlight the potential molecular therapeutic targets for managing and treating chronic rhinosinusitis.

## 1. Introduction

Chronic rhinosinusitis (CRS) can be diagnosed based on the presence of more than two of the following four symptoms for at least 12 weeks: anterior/posterior rhinorrhea, nasal congestion, facial pain or pressure, and olfactory dysfunction. At least one such symptom must be anterior/posterior rhinorrhea or nasal congestion for CRS diagnosis. Depending on the presence or absence of nasal polyps (NPs), CRS can be classified as CRS with NPs (CRSwNP) or CRS without NPs (CRSsNP) [1]. Based on the differences in immune responses between type 1 helper T (Th1)/type 17 helper T (Th17) and type 2 helper T cells (Th2), CRS can also be divided into eosinophilic and non-eosinophilic CRS [2].

CRS affects patients worldwide, with a prevalence rate of 6.95–13% [3,4,5,6]. The burden of medical expenses caused by CRS is exceptionally high. In the United States, the cost of CRS-related diseases is estimated to be between billions and tens of billions of dollars, including outpatient and emergency expenses, drug expenses, nursing expenses, and surgical expenses [7]. Patients with CRS usually receive drug treatment. If the disease persists, the patient can choose to undergo surgery. However, the situation can differ among patients with CRS. Some patients with CRS can be cured by drug treatment only; some patients with CRS require surgery and drug treatment, and others cannot achieve satisfactory curative effects despite trying various treatment schemes [8].

Recently, researchers have made strides to understand different inflammatory mechanisms, which has led to encouraging the development of targeted biological drug therapies. However, owing to the lack of clear treatment guidance and evidence of large-scale use, biologics have not yet become a clinical standard in CRS treatment [9]. Nevertheless, the potential for personalized, targeted therapy for CRS is becoming apparent. Biological therapies are different from traditional therapies. They are aimed at specific immune cells or inflammatory pathways in the disease development and progression. Therefore, they can improve drug efficacy and reduce the incidence of complications [10]. Several biotherapeutics have been developed and approved for patients with CRS. For example, dupilumab is an anti-interleukin (IL)4Rα antagonist that prevents IL-4 and IL-13 from binding to their receptors and blocks the downstream signal transduction of the Th2 inflammatory pathway, and can be used to treat refractory CRSwNP [11].

Based on the increasingly urgent need for new treatment strategies for patients with CRS, remarkable progress has been made in investigating and developing biological therapies. This review summarizes the current and in-development therapeutic strategies of biologics for CRS treatment, and it aims to provide doctors in relevant fields with the latest information concerning biologics.

## 2. Biologics in CRSwNP

Most biological agents for CRSwNP were originally biological agents for asthma because the pathophysiology of CRSwNP is similar to that of asthma [12]. Biological agents approved by the US Food and Drug Administration (FDA) for CRS include dupilumab, omalizumab, and mepolizumab, whereas benralizumab and reslizumab are in clinical trials [13]. Table 1 lists the names and brands of commercial biological agents approved or in clinical trials for the treatment of CRSwNP, the reduced biomarkers in subjects after their treatment, and their common side effects (Table 1).

### 2.1. Dupilumab

IL-4 and IL-13 are key cytokines involved in the pathogenesis of CRSwNP [14]. IL-4 promotes the differentiation of Th0 cells into Th2 cells and is involved in mast cell activation [15]. IL-13 contributes to eosinophil migration, mucinous gland hyperplasia, and mucus hypersecretion [16]. Because of these effects, these two cytokines play a crucial role in tissue remodeling and NP formation in CRS [17]. Dupilumab (Dupixent^®^) is a humanized monoclonal antibody that targets the IL-4 receptor α-subunit, which is shared by IL-4 and IL-13 receptors. It effectively regulates Th2 inflammation by inhibiting IL-4 and IL-13 signal transduction. The dual inhibition of IL-4 and IL-13 signaling may be an essential strategy for treating CRSwNP [18]. The FDA approved dupilumab for moderate-to-severe atopic dermatitis (AD) and moderate-to-severe asthma in 2017 and 2018, respectively. After Bachert et al. [19] published the results of two phase 3 clinical studies, dupilumab was approved for CRSwNP in 2019. Their results showed that, compared with placebo, dupilumab significantly reduced the levels of blood thymus and activation of regulatory chemokines and other cytokines related to Th2 inflammation. Another randomized controlled trial of 724 patients revealed that dupilumab improved the quality of life of patients with CRSwNP complicated by asthma [20]. The experimental group with a subcutaneous injection of 300 mg dupilumab every 2 weeks had better nasal congestion scores, Lund Mackay computed tomography scores, and 22-item sinonasal outcome test scores than the control group with placebo. The scores for 6-item asthma control were also significantly improved. As the first biotherapeutic drug approved by the FDA for CRSwNP, it can be used for patients with routine treatment failure, but it causes a certain economic burden for patients with CRSwNP because of its high price. As a biological agent approved for clinical use in recent years, the side effects of dupilumab have attracted attention. An observational study indicated that dupilumab is also related to herpes simplex virus reactivation and conjunctivitis, and common drug injection site reactions [21].

### 2.2. Omalizumab

Immunoglobulin E (IgE) is an effective inflammatory mediator that activates mast cells and aggravates inflammatory signal transduction [22]. When the human body is exposed to allergens, IgE is produced in large quantities. However, the specific role of IgE in the pathophysiology of CRSwNP remains unclear. Elevated IgE levels were measured in patients with CRSwNP with AD and eosinophilia [23]. Omalizumab is a monoclonal anti-IgE antibody that can prevent IgE from interacting with receptors on the cell surface and prevent the activation of basophils and mast cells by binding to Fc receptors on various inflammatory cells [24]. It is sold under the brand name Xolair^®^ and was originally approved by the FDA to treat chronic idiopathic urticaria and allergic asthma [25,26]. The pathophysiological mechanism related to the production of local intranasal IgE leads to an inflammatory cascade and perhaps to the occurrence of CRSwNP [27]. The results of two randomized phase 3 clinical trials showed that omalizumab significantly improved the indicators and symptoms of patients with severe CRSwNP with insufficient intranasal corticosteroid response and was officially approved by the FDA in 2020 for the treatment of adult refractory CRSwNP [28]. Bidder et al. evaluated the efficacy of omalizumab in patients with CRSwNP complicated with asthma in a 16-week study and found that the nasal symptoms of the experimental group were improved compared with those of the control group [29]. A retrospective study showed that serum IgE levels and tissue eosinophils did not decrease in patients with CRSwNP after omalizumab [30]. Similarly, another study showed that treatment with omalizumab did not reduce total IgE levels in patients with CRSwNP and asthma [31]. Only one study has reported that omalizumab reduced serum periostin levels in patients with CRSwNP [32]. The primary adverse reaction to omalizumab is injection site reactions [33]. Other adverse reactions, including anaphylaxis and headaches, are like those associated with other biological agents [34]. The economic burden of omalizumab was the same as that of dupilumab. Patients need approximately USD 10,000 to 30,000 for omalizumab treatment every year, which is difficult for most patients to bear [35].

### 2.3. Mepolizumab

IL-5, produced by group 2 innate lymphoid cells (ILC2s) or mast cells, plays an important role in Th2 CRSwNP. It mainly involves eosinophilia and plasma cell activation in NPS [36]. Mepolizumab (Nucala^®^) is an anti-IL-5 monoclonal antibody. Several studies have reported the effects of drugs on CRSwNP. It inhibits eosinophilic inflammation by interfering with the binding of IL-5 to receptors expressed on eosinophils and basophils [37]. In 2015, the FDA first approved mepolizumab as an additional maintenance treatment for patients with severe asthma [38]. In 2021, the FDA approved mepolizumab for the treatment of CRSwNP. It was the first anti-IL-5 biological agent approved by the FDA for adult patients with CRSwNP. A 2011 study evaluated the efficacy of mepolizumab in 30 patients with severe CRSwNP, all of whom showed poor drug control. The patients were treated twice with 1500 mg of mepolizumab for 8 weeks. Compared to the control group treated with placebo, the nasal polyp size of patients in the mepolizumab treatment group was significantly reduced, and the score was significantly improved, showing a good curative effect [39]. Another phase 2 clinical trial in 2017 evaluated whether 105 patients with CRSwNP requiring surgery could avoid surgery after receiving mepolizumab. All patients received 750 mg mepolizumab every 4 weeks for 25 weeks. Although the symptoms of sinusitis and the size of NPs in the drug treatment group were reduced compared to those in the control group, 70% of patients still needed surgical treatment [40]. A phase 3 clinical study of mepolizumab was conducted between May 2017 and December 2018. A total of 407 patients with CRSwNP [41] were included. Compared with the control group, treatment with mepolizumab reduced the size of NPs and nasal congestion symptoms, showing the possibility of its use in patients with refractory severe chronic sinusitis with NPs. Common adverse reactions of mepolizumab include headache and injection site reactions, and there are no reported cases of allergic reactions. Although several studies have reported death cases, there is no strong evidence of a causal relationship with the use of mepolizumab [42]. The efficacy of mepolizumab is good and is associated with fewer side effects. However, its cost-effectiveness exceeded the value threshold. To reach these thresholds, a price discount of more than 60% from the current list price is required. Thus, relevant personnel must negotiate with drug manufacturers; otherwise, it is difficult to apply to most patients [43].

### 2.4. Reslizumab

Reslizumab is also a humanized anti-IL-5 antibody that can prevent IL-5 from binding to IL-5 receptors and reduce eosinophil differentiation, similar to mepolizumab [44]. Reslizumab (CINQAIR^®^) was approved in the United States as an additional maintenance treatment for severe asthma with an eosinophilic phenotype in 2016. However, it has not been approved for the treatment of CRSwNP [45]. Gevaert et al. evaluated the safety, efficacy, and biological activity of reslizumab using parameters such as NP size, sinusitis symptoms, and local IL-5 levels. Their results showed that a 3 mg/kg dose of reslizumab was completely safe in a single injection. A single reslizumab injection improved the nasal polyp score in half of the patients with CRSwNP, and the number of eosinophils in the serum and nasal secretions decreased significantly within 8 weeks after treatment. Although there has been no trial of reslizumab for nasal polyposis, a phase III trial of reslizumab for CRS is currently in the recruitment stage [46].

### 2.5. Benralizumab

Benralizumab is a monoclonal antibody that binds to the IL-5 receptor on the surface of eosinophils to activate the anti-IL-5 receptor α-subunit. It binds to the IL-5 receptor, resulting in signal transduction degradation and strong apoptosis of eosinophils [47]. Benralizumab (Fasenra^®^) was approved by the FDA to treat severe asthma in 2017 and has been approved in the European Union, Canada, Australia, and several other countries [48]. The results of a phase III-B clinical trial involving 153 patients with CRSwNP by Canonica et al. showed that benralizumab significantly improved nasal symptoms, and other clinical symptoms related to CRSwNP improved to varying degrees without changing over time [18]. A randomized controlled trial evaluating benralizumab in the treatment of CRSwNP involving more than 300 people is currently underway [49], and a phase III clinical trial of benralizumab in the treatment of severe nasal polyposis is currently recruiting subjects. These results indicate that benralizumab for CRSwNP is expected to obtain FDA approval over the next few years. This will make benralizumab a new CRS biological agent after dupilumab, omalizumab, and mepolizumab; reslizumab will compete for this position.

### 2.6. Anti-IL-25/33 and Anti-TSLP

Thymic stromal lymphopoietin (TSLP), IL-25, and IL-33 are key regulators of the immune pathogenesis of CRS [50]. The activation of epithelial cells stimulated by allergens and pathogens releases TSLP, IL-25, and IL-33, resulting in the activation of Th2, ILC2, and dendritic cells (DCs). The overproduction of inflammatory factors, such as IL-4, IL-5, and IL-13, from these cells leads to an increase in tissue eosinophils and the production of local IgE (Figure 1). In addition to the above biological agents targeting IL-4, IL-5, IL-13, and IgE, biological agents targeting IL-25, IL-33, and TSLP have also been studied. Shin et al. reduced the number of polyps, mucosal edema thickness, and inflammatory cell infiltration through anti-IL-25 treatment and inhibited the expression of local inflammatory cytokines, such as IL-4 and interferon (IFN)-γ, indicating the possibility of treating NPs with anti-IL-25 therapy [51]. IL-33 plays a role in the pathogenesis of allergic diseases and functions as a neutrophil activator. The monoclonal antibody etokimab targeting IL-33 will be evaluated in the upcoming phase 2 trials for adult controlled trials with CRSwNP [52]. Tezepelumab is a monoclonal antibody targeting TSLP. In a phase 2 clinical trial, after 52 weeks of treatment, the efficacy of subcutaneous injection of tezepelumab was compared with that of the control group. Tezepelumab can significantly reduce the incidence of asthma in patients and has considerable potential as a therapeutic agent for CRSwNP [53].

As shown above, biological agents have shown potential for the treatment of CRSwNP. Currently, three biological agents have been approved by the FDA for refractory CRSwNP, and an increasing number of biological agents are undergoing clinical trials. Although several influential clinical trials have proven preliminary efficacy and safety, when the treatment time is prolonged, the new treatment has a high risk of unknown long-term side effects.

### 2.7. Uncontrolled CRSwNP

At the European Forum for Research and Education in Allergy and Airway Diseases (EUFOREA) held in Belgium in 2020 [54], the committee of experts defined uncontrolled CRSwNP as “persistent or recurrent despite long-term intranasal steroid treatment and at least one course of systemic corticosteroid treatment and/or nasal surgery in the past two years”. Current research has not shown that biological agents can replace surgical therapies. Therefore, biological agents should be used as another treatment for uncontrolled CRSwNP or as an option for patients who cannot undergo surgical treatment rather than completely replacing the traditional treatment. The EUFOREA expert group proposed five criteria: evidence of type 2 inflammation, the need for systemic corticosteroids in the past two years, significant impairment of quality of life, significant loss of smell, and the diagnosis of asthma. Their research concluded that biological agents like dupilumab, omalizumab, and benralizumab are suitable for patients with bilateral nasal polyps who have previously undergone sinus surgery and meet the above three criteria. However, if patients have never received surgical treatment, they need to meet at least the above four criteria before they can consider receiving biological treatment [55]. The approach based on biomarker-defined clusters would identify responders to biologics [27], for example, in a study of the anti-IL-5 biological agent reslizumab, it was found that in the patients with bilateral nasal polyps, only elevated IL-5 levels (>40 pg/mL) in nasal secretions before treatment benefited from anti-IL-5 treatment [56]. Different biologics should be selected according to the molecular markers of each patient, including anti-IL-4 receptor α (dupilumab), anti-IgE (omalizumab), and anti-IL-5/RA (mepolizumab and benralizumab).

## 3. Biologics in CRSsNP

Although many studies on the efficacy of biological agents have been conducted in CRSwNP, the exploration of the efficacy of biological agents in CRSsNP is still in the theoretical stage, lacking attention and research. CRSsNP differs pathophysiologically from asthma, which is a type 2 inflammatory disease; instead, CRSsNP is a type 1 inflammatory disease [57]. Therefore, CRSsNPs are generally not considered the focus of type 2 targeted biological agents. However, increasing evidence indicates that it may be wrong to roughly classify CRSwNP and CRSsNP into type 2 and non-type 2 inflammatory diseases. In the United States and Europe, the most common inflammatory endotype of CRSwNP is type 2 CRSwNP. However, in Asia, patients with CRSwNP in some regions mainly show a mixed pattern of type 2 and non-type 2, whereas patients with CRSwNP in other regions show lower Th2 expression levels [58,59]. At present, no biotherapy for non-type 2 mediated CRS inflammation is under study; however, the above research results suggest that if biological agents for non-type 2 CRS can be developed, it may be a meaningful achievement for many patients with CRS in Asia.

Figure 2 shows various biological therapies for non-type 2 inflammatory diseases, which may be applied to CRSsNP in the future. The biological agent ustekinumab, anti-IL-12/23, has shown good efficacy in joint pain and psoriatic arthritis in clinical studies involving more than 150 patients. It showed high sensitivity to pyoderma gangrenosum and erythema nodosum in seven studies, including 65 patients [60]. In a controlled study involving 230 patients with multiple sclerosis, the lesion site in the daclizumab group targeting IL-2R was more than 70% less than that in the placebo group. The incidence of side effects in the two groups was the same, such as skin rash and infection [61]. A randomized study evaluated the safety and efficacy of fontolizumab, an antibody targeting IFN-γ, in 135 patients with Crohn’s disease. The results showed that the C-reactive protein level in the treatment group decreased significantly and was well tolerated without severe side effects [62]. Canakinumab is a biological agent targeting IL-1 β. In a clinical study evaluating whether canakinumab can be used to treat autoimmune recurrent fever syndrome, the results showed that canakinumab could effectively control and prevent attacks in patients with various autoimmune diseases, with an effective rate of more than 70% [63]. A phase III clinical study reported the efficacy of sirukumab targeting IL-6 in treating rheumatoid arthritis. This 2-year study involving more than 1000 patients showed that sirukumab treatment reduced the clinical signs and symptoms of patients with rheumatoid arthritis and minimized the progress of radiographic damage [64]. Ixekizumab, an antagonist of IL-17A, can disrupt the inflammatory cascade in psoriasis [65]. The phase III clinical trial results showed that nearly 90% of patients treated with ixekizumab for 12 weeks showed efficacy, and the area severity index of psoriasis decreased by 75% [66]. The efficacy and safety of fezakinumab (an IL-22 monoclonal antibody) in patients with moderate-to-severe AD were evaluated in a phase 2 clinical trial [67]. After 10 weeks of treatment, the AD score of patients treated with fezakinumab was lower than that of patients treated with the placebo. None of the above drugs have been clinically evaluated in patients with CRS, but these studies may lead to studies related to non-type 2 CRS.

## 4. Future Targets of Biologics in CRS

Just as the application of biological agents in CRSwNP is inspired by the use of biological agents in asthma [68], research on biological agents for Th1 and Th17 inflammatory pathway-mediated CRS will also be inspired by the evaluation or approval for the treatment of inflammatory diseases with overlapping pathophysiology with CRS, such as psoriatic arthritis, Crohn’s disease, and AD. The main inflammatory pathways related to CRS include the Th1, Th2, and Th17 mediated inflammatory pathways [69], and different targets in these pathways will become the focus of research on new biological agents targeting CRS in the future.

IFN-γ inducible protein 10 (CXCL10) is produced by IFN-γ chemokines induced and secreted by monocytes, neutrophils, fibroblasts, DCs, and other cells [70]. CXCL10 expression level is elevated in different diseases such as rheumatoid arthritis and psoriatic arthritis [71,72]. CXCL10 is a receptor-binding ligand of the chemokine receptor CXCR3, and the CXCL10–CXCR3 axis is a pathway for IFN-γ induction. Therefore, CXCL10 and CXCR3 may play essential roles in leukocyte homing to inflammatory tissues and the persistence of inflammation [73]. In the IFN γ-induced vitiligo mouse model, mice with identified depigmentation treated with CXCL10 neutralizing antibody for 8 weeks showed reversal of the disease through pigmentation [74]. Elevated IFN-γ level is one of the disease manifestations of CRS [75], and the increase of CXCL10 in CRS can be predicted, which is also supported by the research results. The researchers’ results indicated that the expression levels of CXCL10 mRNA and protein in nasal fibroblasts of patients with type 2 CRS were significantly enhanced compared to that in normal nasal mucosal cells [76]. Although there are no clinical trials related to anti-CXCL10 biologics in patients with CRS, CXCL10 has the potential to become a target of biological agents for CRS treatment in the future.

The tumor necrosis factor receptor OX40 (CD134) is activated by its cognate ligand OX40L (CD134L, CD252) and co-stimulates T cells [77]. The OX40L-OX40 axis participates in the continuous activation of effector T cells and can accelerate the differentiation of Th1 and Th2 effector cells [78]. Therefore, the inhibition of OX40 may have therapeutic significance in T cell-mediated diseases. A phase 2 clinical study investigated the safety and efficacy of antagonistic antibodies against OX40 in patients with AD. Their study found that after 71 days of injection, the treatment group achieved more than 50% improvement in eczema severity than the placebo group and showed good tolerance, highlighting the potential of OX40 targeted treatment of AD patients [79]. A study using an allergic rhinitis (AR) mouse model also confirmed the potential of OX40 as a therapeutic target for AR. After blocking the OX40/OX40L signal pathway through small interfering RNA (siRNA) interference, researchers found that it can inhibit the allergic reaction in AR mice and effectively alleviate allergic symptoms, such as nose scratching, sneezing, and runny nose [80]. As for CRSwNP, experimental results in the superior nasal cortex of 94 NPs tissues showed that a higher expression level of OX40L was detected in the DCs of NPs nasal mucosa [81], indicating that OX40L may play an important role in the pathogenesis of CRSwNP. If relevant research is conducted in the future, the anti-OX40 biologics may become one of the biologics for the treatment of CRS in the future.

## 5. Conclusions

Based on an in-depth understanding of the primary drivers of inflammation in CRS, multiple therapeutic targets have been identified to match the appropriate biological agents for CRS treatment. There is considerable clinical evidence supporting the safety and efficacy of biologics in patients with CRS, and the FDA has approved several such drugs for treating CRS. However, the high cost of drugs makes it difficult for people to access such treatment options. In the treatment of patients with uncontrolled CRSwNP, after evaluating the efficacy of surgery and biological agents, clinicians choose sinus surgery, biological methods, or a combination of both, which brings high utility benefits. Using biomarkers to classify and identify different recipients of biologics will also lead to more accurate treatment options. As more studies focus on new therapeutic targets and accurately matched biological agents based on CRS endotypes, the application range of biological agents in various CRS endotypes will be expanded, and the therapeutic effects of biological agents will also be improved. However, it is undeniable that the application of precision medicine and lowering of the associated cost will be essential to ensure that biological agents can become a new treatment option for most patients in the future, in addition to traditional drugs and surgery.

## Figures and Tables

**Figure 1 ijms-23-05523-f001:**
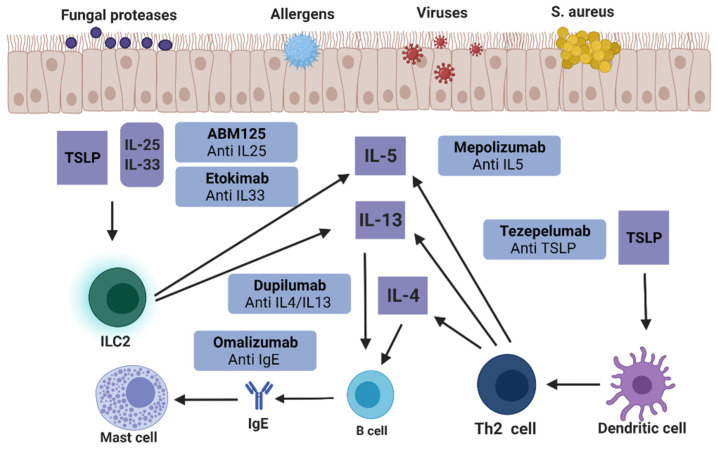
Biologics for treating CRSwNP.

**Figure 2 ijms-23-05523-f002:**
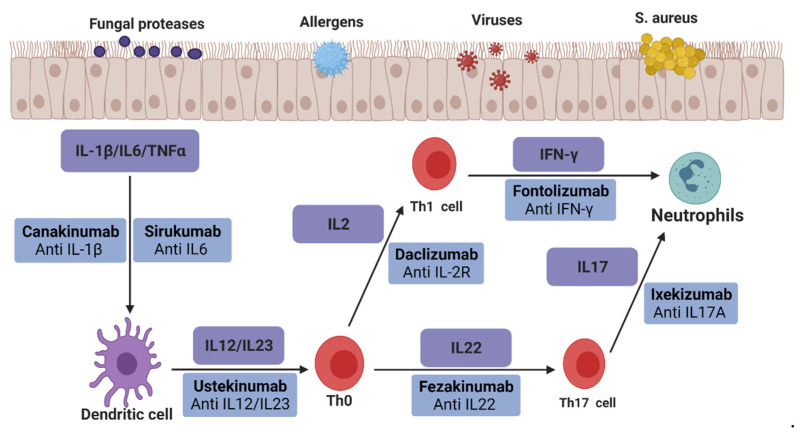
Possible biologics for treating CRSsNP.

**Table 1 ijms-23-05523-t001:** Biological agents approved or in clinical trials for CRSwNP.

Target	Drug	Brand	Approval or Development Status for CRS	Decreased Biomarker after Treatment	Side Effects
IL-4Rα	Dupilumab	Dupixent^®^	FDA approved for CRSwNP	Blood thymus and activation-regulated chemokine	Injection site reaction and conjunctivitis
IgE	Omalizumab	Xolair^®^	FDA approved for CRSwNP	Blood Periostin	Injection site reactions
IL-5	Mepolizumab	Nucala^®^	FDA approved for CRSwNP	Blood Eosinophil	Headache and injection site reaction
IL-5	Reslizumab	CINQAIR^®^	Phase 2 trials concluded	Blood Eosinophil	Nasopharyngitis
IL-5Rα	Benralizumab	Fasenra^®^	Phase 3 trials concluded	Blood Eosinophil	Nasopharyngitis

## Data Availability

Not applicable.

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
