# Peer review of "Therapeutic Strategies of Biologics in Chronic Rhinosinusitis: Current Options and Future Targets"

_ijms, 2022, doi:10.3390/ijms23105523_

Round 1
Reviewer 1 Report
Se trata de un trabajo de revisión que recoge de manera concisa y ordenada la evidencia actual del papel de los fármacos biológicos en la CRSwNP de tipo 2.
It also posits the future of biologic therapies for non-type 2 CRSwNP. This gives the reader a perspective that is not often the case when writing biologics for nasal polyposis. And in this way the possibilities of treatment for the Asian endotype are deepened.
On the other hand, it makes a brief position on the role of surgery and the rest of current treatments for CRSwNP based on the best current evidence, giving it a leading role that should not be missed.
Author Response
Thank you very much for your comments and careful review of our manuscript.
Sincerely,
Tae Hoon Kim M.D., PhD. (corresponding author)
Reviewer 2 Report
The manuscript makes a "pharmacological" review of biological drugs for the treatment of chronic rhinosinusitis, focusing on Th2 inflammation. In this aspect, it brings few novelties.
I believe that the authors should also delve into the definition of "uncontrolled CRSwNP" and, therefore, into the indications for biological drugs in this disease. Likewise, define the criteria of response to treatment and the therapeutic indication based on the molecular markers of each patient: interleukins, IgE... This last aspect is "named" in the conclusions, but a slightly deeper analysis is required.
Author Response
We thank the reviewers for their constructive suggestions, which helped us revise the manuscript. A point-by-point response is attached below.
Black: Reviewer’s comment
Red: Author’s response
Blue: Text in the revised manuscript (line number)
Point 1: I believe that the authors should also delve into the definition of "uncontrolled CRSwNP" and, therefore, into the indications for biological drugs in this disease. Likewise, define the criteria of response to treatment and the therapeutic indication based on the molecular markers of each patient: interleukins, IgE... This last aspect is "named" in the conclusions, but a slightly deeper analysis is required.
Response 1: Thank you for your comments. This part is really the part that this article lacks. A small chapter has been added to the manuscript. And a sentence is also added to the conclusion.
Line 204:
2.7. Uncontrolled CRSwNP
At the European Forum for Research and Education in Allergy and Airway Diseases (EUFOREA) held in Belgium in 2020 [56], the committee of experts defined uncontrolled CRSwNP as "persistent or recurrent despite long-term intranasal steroid treatment and at least one course of systemic corticosteroid treatment and/or nasal surgery in the past two years". Current research has not shown that biological agents can replace surgical therapies. Therefore, biological agents should be used as another treatment for uncontrolled CRSwNP or as an option for patients who cannot undergo surgical treatment rather than completely replacing the traditional treatment. The EUFOREA expert group proposed five criteria: evidence of type 2 inflammation, the need for systemic corticosteroids in the past two years, significant impairment of quality of life, significant loss of smell, and the diagnosis of asthma. Their research concluded that biological agents like dupilumab, omalizumab and benralizumab are suitable for patients with bilateral nasal polyps who have previously undergone sinus surgery and meet the above three criteria. However, if patients have never received surgical treatment, they need to meet at least the above four criteria before they can consider receiving biological treatment [57]. The approach based on biomarker-defined clusters would identify responders to biologics [28], for example, in a study of the anti-IL-5 biological agent reslizumab, it was found that in the patients with bilateral nasal polyps, only elevated IL-5 levels (> 40 pg/ml) in nasal secretions before treatment benefited from anti-IL-5 treatment [58]. Different biologics should be selected according to the molecular markers of each patient, including anti-IL-4 recep-tor α (dupilumab), anti-IgE (omalizumab), and anti-IL-5 / RA (mepolizumab and benrali-zumab).
Line 322:
In the treatment of patients with uncontrolled CRSwNP, after evaluating the efficacy of surgery and biological agents, clinicians choose sinus surgery, biological methods, or a combination of both, which brings high utility benefits. Using biomarkers to classify and identify different recipients of biologics will also lead to more accurate treatment options.
Sincerely,
Tae Hoon Kim M.D., PhD. (corresponding author)
Round 2
Reviewer 2 Report
I thank the authors who have introduced the suggested changes and clarifications and the manuscript has substantially improved.